# Cell salvage in bacterially contaminated surgical fields – A scoping review

Joeri Slob[1º], Pim Hondebrink[2º], Ankie W.M.M. Koopman – van Gemert[3],
Margriet E. van Baar[1,4], Cornelis H. van der Vlies[1,5,6], Seppe S. H. A. Koopman[3*]

1 Alliance of Dutch Burn Care (ADBC), Burn Center, Maasstad Hospital, Rotterdam, The Netherlands,
2 Faculty of Medicine, Erasmus MC, University Medical Center Rotterdam, Rotterdam, The Netherlands,
3 Department of Anaesthesiology, Maasstad Hospital, Rotterdam, The Netherlands, 4 Department
of Public Health, Erasmus MC, University Medical Center Rotterdam, Rotterdam, The Netherlands,
5 Department of Surgery, Trauma Research Unit Department of Surgery, Erasmus MC, University Medical
Center Rotterdam, Rotterdam, The Netherlands, 6 Department of Trauma and Burn Surgery, Maasstad
Hospital, Rotterdam, The Netherlands

º These authors contributed equally to this work.
* koopmanJ@maasstadziekenhuis.nl

## Abstract

### Objective

This scoping review systematically examines the use of intraoperative cell salvage in bacterially contaminated surgical settings, examining contamination levels, decontamination strategies, and clinical outcomes following the reinfusion of contaminated salvaged blood.

### Methods

Following the Joanna Briggs Institute methodology and the PRISMA extension for scoping reviews, comprehensive searches were conducted in MEDLINE, Embase, Web of Science, and Cochrane CENTRAL. Eligible studies included primary empirical research reporting bacterial contamination of salvaged blood confirmed by culture, as well as studies reporting on decontamination strategies or clinical outcomes following reinfusion, across any surgical setting. Data was extracted and synthesized descriptively. Key findings were summarized in structured tables.

### Results

Thirty-nine studies, involving 1654 patient, met the inclusion criteria. Bacterial contamination was consistently reported, with positive culture rates varying widely. Methodological quality was assessed for all studies, except the ex-vivo studies. Decontamination strategies, including leukocyte depletion filtration (10 studies) and antibiotic additives (4 studies), reduced contamination, though effectiveness varied. Among 26 studies, involving 1203 patients, who received reinfused salvaged blood,

**Data availability statement:** All relevant data are within the manuscript and its Supporting Information files.

**Funding:** This work is supported by the Dutch Burns Foundation (grant no. 19.102). The funder had no influence on the design or content of this manuscript. There was no additional internal or external funding received for this study.

**Competing interests:** The authors have declared that no competing interests exist.

postoperative infections were reported in 15 studies. A total of 55 cases of postoperative bacteraemia were identified. Only five studies described a plausible microbiological link between the reinfused blood and subsequent infection, involving nine patients in whom the isolated pathogen matched the organism cultured from the salvaged blood.

## Conclusion

Bacterial contamination of salvaged blood occurred frequently, even in procedures not typically classified as contaminated. Decontamination strategies demonstrated variable effectiveness in reducing bacterial contamination. Despite contamination, a potential microbiological link between reinfused salvaged blood and infection was described in only nine patients across five studies. However, substantial heterogeneity in methodologies and a small sample size of most studies makes it difficult to draw definite conclusions about safety. Therefore, we would advise to use cell salvage in known bacterial contaminated areas on a case by case basis, after careful evaluation of the pros and cons, until future research defined safe contamination thresholds, evaluated the effectiveness of decontamination techniques, and assessed clinical outcomes in standardized and controlled settings.

## Introduction

Intraoperative cell salvage (ICS) is a technique in which autologous, shed blood from the surgical field is collected, processed, and reinfused into the patient. Cell salvage was developed as a way to minimize allogeneic transfusion requirements. By reinfusing the patient's own blood, ICS reduces exposure to allogeneic blood and thereby helps to avoid transfusion-related risks such as immunologic reactions [1–4]. Currently, ICS is widely utilized across different surgical specialties, including cardiac, orthopaedic and trauma surgery, as part of blood conservation strategies [5–10]. A recent Cochrane review confirms that in certain elective surgeries, the use of ICS significantly decreases the need for allogeneic blood transfusions without increasing adverse outcomes [10].

Despite its potential benefits, the application of ICS in bacterially contaminated surgical fields remains controversial. Both manufacturers and clinical guidelines suggest caution in contaminated settings. Manufacturers of ICS devices list active infection or gross contamination as contraindications due to the risk of bacteraemia [11,12]. Similarly, international guidelines, for example from the UK Cell Salvage Action Group, generally discourage ICS in infected settings [13]. Simultaneously, guidelines suggest that ICS may be considered in selected cases. For instance, the Association of Anaesthetists guidelines (ASA) state that the use of ICS in infected or contaminated surgical fields should be assessed on a case-by-case basis, as there is no conclusive evidence that ICS leads to poorer clinical outcomes [14].

ASA also advises the use of a leukocyte depletion filter (LDF) when ICS is applied in a surgical field that might be contaminated. LDFs have been shown to significantly

reduce, and in some cases even eliminate bacterial contamination [15,16]. However, the exact mechanism by which LDFs reduce bacterial load is not fully understood, as bacteria are typically small enough to pass the filter pores. It is thought that bacterial removal occurs through adhesive interactions between bacterial cell surfaces and the filter material [17].

While these findings are encouraging, conclusive evidence regarding the safety of ICS in contaminated surgical fields is still lacking. There is no consensus on what level of bacterial contamination in salvaged blood is safe for reinfusion, or which decontamination protocols are most effective. Most published studies are small or context-specific, and their methodologies vary widely. As a result, clinicians are often required to make case-by-case decisions on using ICS in contaminated surgical fields, weighing the benefits of autologous transfusion against potential infection risks. To date, no review has comprehensively mapped the literature on ICS use in the context of bacterial contamination. Given the broad scope and methodological heterogeneity of the current evidence base, we conducted a scoping review to systematically assess existing evidence.

## Methods

This review aims to assess: 1) bacterial contamination: characterize the types and levels of bacterial contamination encountered during ICS; 2) reduction strategies: describe the techniques used to reduce or manage contamination during salvage and reinfusion; 3) clinical outcome: document the reported clinical outcomes following reinfusion of bacterially contaminated salvaged blood; and 4) knowledge gaps: identify key uncertainties to inform future clinical studies or guideline development.

This scoping review was conducted in accordance with the methodological framework developed by the Joanna Briggs Institute (JBI) [18] and is reported in line with the PRISMA extension for Scoping Reviews [19]. An a priori protocol was developed following JBI guidelines [20], structured using the JBI template for scoping reviews [21], and registered in PROSPERO database (CRD420251023866).

### Search strategy

A comprehensive search strategy was developed in collaboration with an experienced information specialist from the Erasmus MC Medical Library. Four electronic databases were systematically searched: MEDLINE, Embase, Web of Science Core Collection, and Cochrane CENTRAL (S1 Appendix). Search strings were tailored to each database and included terms related to intraoperative cell salvage, bacterial contamination, and clinical outcomes. No date restrictions were applied, and only studies published in English or Dutch were considered. Reference lists of relevant reviews identified during the screening process and of included articles were also checked to identify additional eligible publications.

### Eligibility criteria

Eligibility criteria were defined using the Population-Concept-Context (PCC) framework [18]. Eligible studies included studies involving human subjects of any age or patient population, or animal models undergoing surgical procedures with the use of ICS. Additionally, preclinical models that simulate clinical use of ICS and assess bacterial contamination and/or reduction techniques were included.

Studies had to examine or describe at least one of the following: measurement of bacterial contamination in salvaged blood, confirmed by blood cultures; bacterial reduction strategies implemented during the cell salvage process, such as leukocyte depletion filtration or antibiotic treatment of salvaged blood; or evaluation of clinical outcomes following reinfusion of bacterially contaminated salvaged blood.

Studies were excluded if they did not quantify or confirm bacterial contamination of salvaged blood or failed to provide sufficient detail to determine whether such contamination was assessed. Studies focusing exclusively on non-bacterial contamination (e.g., viral or fungal) or solely on the salvage or transfusion of blood components other than red blood cells were also excluded.

 

All surgical settings were eligible for inclusion, provided that bacterial contamination was confirmed by blood cultures and clearly reported. Only primary empirical research was included. Case reports, reviews, editorials, commentaries, clinical guidelines, protocols, and conference abstracts were excluded from this review.

### Study selection

Search results were imported into Covidence systematic review software (Veritas Health Innovation, Melbourne, Australia). Two independent reviewers (PH, JS) screened titles and abstracts against the eligibility criteria. Full texts of potentially relevant studies were then assessed independently by the same reviewers. Discrepancies were resolved through discussion or consultation with a third reviewer (AK). If full-text articles were not readily accessible, attempts were made to obtain them through institutional library services. Reasons for exclusion at the full-text stage were recorded. The selection process was documented in a PRISMA-ScR flow diagram.

### Data extraction

A standardized and piloted data extraction form was used to chart data from included studies. One reviewer (PH) extracted the data, and a second reviewer (JS) verified the accuracy and completeness of the extracted data.

### Methodological quality

The quality assessment of articles included was carried out by JS and PH independently. Whenever applicable a third reviewer (AK) arbitrated. The revised Cochrane risk of bias 2 tool for randomised controlled trials (RoB 2) was used for the included randomised controlled trials [22]. This tool uses stratification into five domains to detect potential bias. For included articles other than randomised controlled trials the Newcastle-Ottawa Score was used [23]. This tool uses stratification into three domains to score quality. Ex-vivo studies were not assessed for methodological quality.

### Data analysis and presentation

Data were synthesized descriptively, with a focus on bacterial contamination, reduction strategies, and clinical outcomes. Findings were summarized in structured tables corresponding to these domains. A narrative synthesis was used to contextualize the data, highlight key patterns, and identify gaps in the current evidence. To enhance clarity and comparability, results were organized by surgical specialty.

### Results

The initial database search (April 2025) yielded a total of 2139 records. Of these, 988 duplicates were automatically removed during the import process, and 5 more duplicates were excluded manually or by Covidence systematic review software. An additional 6 records were identified through reference list checking and expert consultation. Subsequently, 1152 titles and abstracts were screened for eligibility based on previously stated inclusion and exclusion criteria. Following this initial screening phase, 83 full-text articles were retrieved and assessed for inclusion. Of these, 39 studies met the eligibility criteria and were included in the final scoping review. The complete overview of the study identification and selection process is shown in Fig 1.

Table 1 presents the main characteristics of all 39 included studies, with a total of 1654 patients who underwent ICS, grouped by surgical specialty. Table 2 summarizes 14 studies that applied additional decontamination techniques beyond standard ICS processing, including leukocyte depletion filtration (n = 10), antibiotic treatment of salvaged blood (n = 4), and use of a customised antibacterial membrane (n = 1). Table 3 outlines 26 studies reporting reinfusion of salvaged blood in 1203 patients, along with corresponding clinical outcomes. Table 4 and Fig 2 summarize the quality assessments of RCTs (n = 4) and cohort studies (n = 31), except for the ex-vivo studies. The RCTs were assessed as some concerns (n = 4). The

**Cell Salvage in Bacterial Contaminated Surgical Fields**

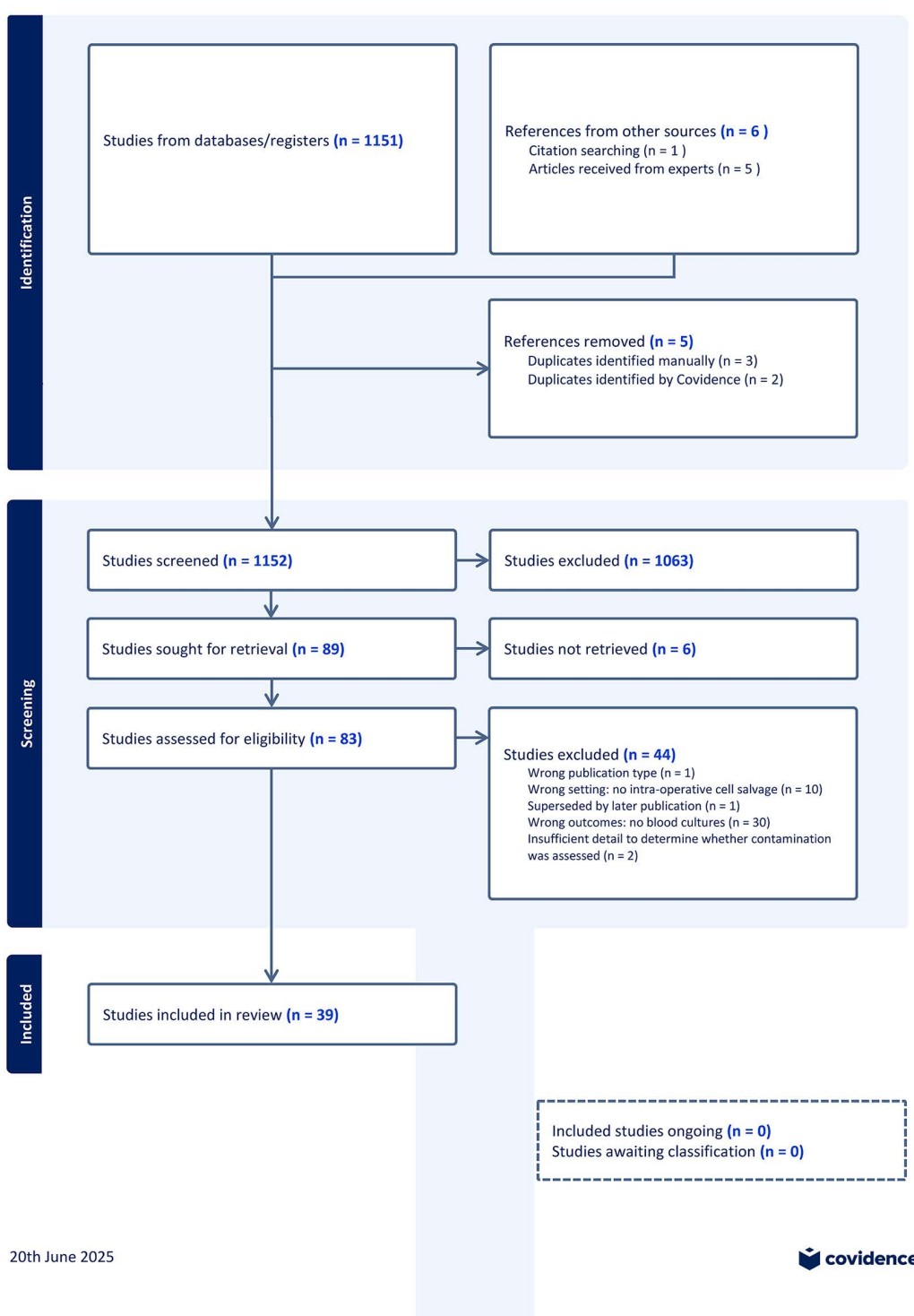

**Fig 1. PRISMA flowchart of study selection process.**

**Table 1. Main characteristics and reported contamination rates of included studies.**

| First author | Year | Country | Study design | Type of surgery/setting | Total sample† (ICS group) | ICS device used | Contamination of final product (%) |
|---|---|---|---|---|---|---|---|
| CARDIOVASCULAR SURGERY | | | | | | | |
| Khan | 1975 | UK | Prospective observational study | Thoracic and cardiovascular surgery (not specified) | 22 (22) | Custom manual system (filtration only) | 15/22 (68) |
| Andrews | 1983 | UK | Prospective observational study | Major abdominal vascular surgery | 28 (28) | Custom manual system (filtration only) | 18/18§ (100) |
| Davies | 1987 | Australia | RCT | Aortic aneurysm and aorto-bifemoral grafts | 25 (25) | Sorenson Receptal Auto-transfusion System | 12/25 (48.0) |
| Ezzedine | 1991 | Belgium | Prospective observational study | Cardiac surgery (valve, coronary, complex) | 401 (401) | Autotransfusion System Receptal (Abbott) + IBM-Cobe 2991 Cell Processor (Cobe Medical) | 51/401 (12.7) |
| Bland | 1992 | USA | Prospective observational study | Cardiac surgery with CPB (not specified) | 38 (38) | Cell Saver (Haemonetics) (model not stated) | 30/31§ (96.8) |
| Reents | 1999 | Germany | Prospective observational study | Cardiac surgery (aortic valve replacement) | 10 (10) | Cell Saver 4 (Haemonetics) | 9/10 (90) |
| Shindo | 2004 | Japan | Prospective observational study | Cardiovascular surgery (not specified); comparison CPB (no ICS) vs no-CPB (with ICS); including abdominal surgery | 116 (52) | Cell Saver (Haemonetics) (model not stated) | ICS: 35/52 (67) CPB: 10/64 (16) |
| Ishida | 2006 | Japan | Prospective observational study | Cardiac surgery with and without CPB (CABG, valve replacement, ASD) | 35 (35) | Cell Saver 5 (Haemonetics) | 26/35 (74.2) |
| Schmidt^ | 2009 | Germany | Prospective observational study | Aortic surgery | 6 (6) | Cell Saver 5 (Haemonetics) | 2/6 (33.3) |
| Luque-Oliveros | 2020 | Spain | Cross-sectional observational study | Cardiac surgery with CPB (valve replacement or CABG); median sternotomy | 201 (201) | C.A.T.S. (Fresenius) | 171/201 (85) |
| Zhou | 2023 | China | Prospective observational cohort study | Cardiac surgery with and without CPB (valve, CABG, major aortic) | 204 (204) | Cell Saver Elite (Haemonetics) | 100/204 (49) |
| LIVER SURGERY | | | | | | | |
| Kang | 1991 | USA | Prospective observational study (two-phase) | Orthotopic liver transplantation (OLT) | Phase 1: 14 (14) Phase 2: 11 (11) | Cell Saver 4 (Haemonetics) | Phase 1: Aerobic: 6/14 (42.9) Anaerobic: 2/14 (14.2) Phase 2: 3/11 (27.2) |
| Feltracco | 2007 | Italy | Prospective observational study | Orthotopic liver transplantation (OLT) | 38 (38) | C.A.T.S. (Fresenius) | 26/38 (68.4) |
| Schmidt^ | 2009 | Germany | Prospective observational study | Hemihepatectomy | 6 (6) | Cell Saver 5 (Haemonetics) | 3/6 (50) |
| Liang | 2010 | China | Prospective observational study | Orthotopic liver transplantation (OLT) | 45 (45) | Cell Saver 5 (Haemonetics) | 3/45 (6.7) |

*(Continued)*

**Table 1.** (Continued)

| First author | Year | Country | Study design | Type of surgery/setting | Total sample† (ICS group) | ICS device used | Contamination of final product (%) |
|---|---|---|---|---|---|---|---|
| Kim | 2022 | Republic of Korea | Prospective observational study | Living donor liver transplantation (LDLT) | 29 (29) | Cell Saver 5 (Haemonetics) | 0/29 (0) |
| Kim | 2024 | Republic of Korea | Prospective observational study | Orthotopic liver transplantation (OLT) | 30 (30) | Cell Saver 5 (Haemonetics) | 7/30 (23.3) |
| ORTHOPAEDIC SURGERY | | | | | | | |
| Wollinsky | 1997 | Germany | RCT | Primary hip arthroplasty, cefuroxime (SG) vs no antibiotic prophylaxis (CG) | SG: 20 (20) CG: 20 (20) | Cell Saver III (Haemonetics) or Autotrans (Dideco) | SG: 3/20 (15) CG: 8/20 (40) |
| Nosanchuk | 2001 | USA | Prospective observational study | Elective spinal surgery | 18 (18) | Autovac Blood Processing System (Boehringer Labs) | 19/31* (61) |
| Perez-Ferrer | 2016 | Spain | Prospective observational study | Paediatric posterior spinal fusion surgery | 24 (24) | Cell Saver 5 (Haemonetics) | 13/24 (52.4) |
| Perez-Ferrer | 2017 | Spain | RCT | Paediatric posterior spinal fusion surgery; vancomycin added to CS wash solution (SG) vs normal washing (CG) | SG: 10 (10) CG: 10 (10) | HaemoLite 2+ (Haemonetics) | SG: 0/10 (0) CG: 5/10 (50) |
| Kruger | 2024 | Germany | Retrospective observational study | Septic two-stage hip revision arthroplasty (SG) vs aseptic revision (CG) | SG: 12 (12) CG: 8 (8) | Not specified | SG: 9/12 (75) CG: 5/8 (63) |
| TRAUMA SURGERY | | | | | | | |
| Timberlake | 1988 | USA | Retrospective observational study | Emergency exploratory laparotomy for abdominal trauma | 11 (11) | Cell Saver (Haemonetics) (model not stated) | 11/11 (100) |
| Ozmen | 1992 | USA | Prospective observational cohort study | Emergency exploratory laparotomy for abdominal trauma | 70 (20) | Cell Saver 1 and Cell Saver 3+ (Haemonetics) | 7/43* (16) |
| Bowley | 2006 | South Africa | RCT | Emergency exploratory laparotomy for abdominal trauma | 44 (21) | Cell Saver 4 (Haemonetics) | 11/12§ (92) |
| GYNAECOLOGY | | | | | | | |
| Yamada | 1997 | Japan | Prospective observational study | Abdominal simple total hysterectomy | 40 (40) | HaemoLite 2 (Haemonetics) | 22/33 (66.7) |
| Waters | 2000 | USA | Prospective observational study | Caesarean section | 14 (14) | Cell Saver 5 (Haemonetics) | 7/14 (50) |
| Teare | 2015 | UK | Prospective observational study | Vaginal delivery (SG) compared to caesarean section (CG) | 50 (50) | Cell Saver 5+ (Haemonetics) | SG: 50/50 (100) CG: 20/20 (100) |
| EAR, NOSE AND THROAT SURGERY & ORAL AND MAXILLOFACIAL SURGERY | | | | | | | |
| Locher | 1992 | Switzerland | Prospective observational study | Transoral maxillofacial surgery | 18 (18) | Cell Saver 3+ (Haemonetics) | 18/18 (100) |
| Lenzen | 2006 | Germany | Prospective observational study | Orthognathic (maxillofacial) surgery | 45 (45) | C.A.T.S. (Fresenius) | 0/45 (0) |
| Wasl | 2016 | South Africa | Prospective observational study | Endoscopic angiofibroma resection | 10 (10) | Not specified | 2/10 (20) |

*(Continued)*

**Table 1.** (Continued)

| First author | Year | Country | Study design | Type of surgery/setting | Total sample† (ICS group) | ICS device used | Contamination of final product (%) |
|---|---|---|---|---|---|---|---|
| MIXED/OTHER | | | | | | | |
| Jeng | 1998 | USA | Prospective observational study | Burn excisional surgery | 8 (8) | Cell Saver 4 (Haemonetics) | 8/8 (100) |
| Sugai | 2001 | Japan | Prospective observational study | Autologous transfusion (ICS, ADTP, HAT), surgery type not specified | 287* (30) | Compact Advanced (Dideco) | ICS: 10/30* (33.3) ATDP: 5/104* (4.8) HAT: 3/117* (2.6) |
| Kudo | 2004 | Japan | Prospective observational study | Neurosurgery (brain tumors, aneurysm clipping, AVM resection), craniotomy and transsphenoidal approach | 50 (50) | Cell Saver (Haemonetics) (model not stated) | Total: 14/30 (46.7) Transsphenoidal: 4/4 (100) Craniotomy: 10/26 (38.5) P<0.05 |
| Gigengack | 2021 | Netherlands | Prospective observational study | Burn excisional surgery | 20‡ (20) | Cell Saver 5+ (Haemonetics) | 20/20‡ (100) |
| EX-VIVO STUDIES | | | | | | | |
| Boudreaux | 1983 | USA | Ex vivo experimental study[B] | Cell salvage washing of expired RBCs inoculated with bacteria[B] | 18 (18) | Cell Saver model 15 (Haemonetics) | NR |
| Marks | 1988 | USA | Ex vivo experimental study | Use of antibacterial membrane in RBC and plasma suspension contaminated with feces vs cell washing | NR | Blood Processor 30S (Haemonetics) | NR |
| Waters | 2003 | USA | Ex vivo experimental study | Leucocyte reduction filtration of expired RBCs inoculated with bacteria | 60 (60) | Sequestra 1000 (Medtronic) | NR |
| Yost | 2017 | USA | Preclinical ex-vivo experimental study (two-phase) | Phase 1: leukocyte reduction filtration of porcine RBC in multiple configurations Phase 2: leukocyte reduction filtration of collected postpartum haemorrhage blood | Phase 1: 6 (6) Phase 2: 4 (6) | Custom peristaltic pump circuit | Phase 1: 6/6 (100) Phase 2: 4/4 (100) |
| Hinson | 2020 | USA | Ex vivo experimental study | Cell salvage washing and leukocyte reduction filtration of canine blood | 30 (30) | C.A.T.S. (Fresenius) | NR |

*Note.* ASD: Atrial Septal Defect, ATDP: Autologous Transfusion Donated Preoperatively, AVM: Arteriovenous Malformation, CG: Control Group, CABG: Coronary Artery Bypass Grafting, C.A.T.S.: Continuous Autotransfusion System, CPB: Cardiopulmonary Bypass, HAT: Haemodilution/Autologous Transfusion, ICS: Intraoperative Cell Salvage, LDLT: Living Donor Liver Transplantation, NR: Not Reported, OLT: Orthotopic Liver Transplantation, RBC: Red Blood Cell, RCT: Randomised Controlled Trial, SG: Study Group.

†included for analysis.

*units blood.

‡from 20 different procedures from n = 16 patients.

§ cultures of salvaged blood were only performed in a subset of patients.

^studies discuss two different types of surgery and is therefore mentioned in two categories.

B study also reports a case series on ICS; however bacterial contamination is not confirmed by blood cultures, therefore this case series does not meet the reviews inclusion criteria and results are not extracted.

cohort studies were assessed as poor (n = 28) or good (n = 3). S1 Table lists the bacterial species cultured from the final processed blood product per study. S2 Table summarises studies that assessed bacterial contamination both before and after washing. S3 Table provides quantitative contamination data.

**Table 2. Additional decontamination strategies during cell salvage process.**

| First author | Year | Type of LDF | Additional decontamination | Pre-wash contamination, quantified: | Pre-wash samples contaminated (%) | Post-wash/pre-LDF contamination, quantified: | Post-wash samples contaminated (%) | Post-LDF contamination, quantified: | Post-LDF samples contaminated (%) | Post-other decontamination, quantified: | Post-other decontamination samples contaminated (%): | Notes |
|---|---|---|---|---|---|---|---|---|---|---|---|---|
| CARDIOVASCULAR SURGERY | | | | | | | | | | | | |
| Ezzedine | 1991 | NA | Cefazolin 2g added to each litre of ACD-solution | NR | NR | NR | NR | NA | NA | Positive cultures ≤2 CFU/mL: 42/51 (82%) Positive cultures <5 CFU/mL: 48/51 (94%) | 51/401 (12.7) | No comparator group. |
| LIVER SURGERY | | | | | | | | | | | | |
| Liang | 2010 | FTS-RC202 (Shuangweibio Co.) | No | NR | 28/45 (62.2) | NR | 15/45 (33.3) | NR | 3/45 (6.7) | NA | NA | Reduction Cell Saver + LDF: 25/28, 90.3%, p < 0.001 |
| Kim | 2022 | RCEZIT (Pall Biomedical Products Co.) | No | NR | NR | NR | 13/29 (44.8) before BDA; 9/29 (31.0) after BDA | NR | 0/29 (0) before BDA; 0/29 (0) after BDA | NA | NA | 100% conversion by LDF (before BDA: p < 0.001, after BDA: p = 0.008) |
| Kim | 2024 | RCEZIT (Pall Biomedical Products Co.) | No | NR | NR | NR | 7/30 (23.3) before graft reperfusion; 11/30 (36.7) after graft reperfusion (P = 0.25) | NR | 7/30 (23.3) after graft reperfusion | NA | NA | Conversion ratio LDF (after graft reperfusion) 36.4% (P = 0.25) |
| ORTHOPAEDIC SURGERY | | | | | | | | | | | | |
| Perez-Ferrer | 2017 | NA | Vancomycin (10 µg/mL) added to wash solution | NR | NR | NR | CG: 5/10 (50) SG: 0/10 (0) (p = 0.016) | NA | NA | NR | CG: 5/10 (50) SG: 0/10 (0) (p = 0.016) | |
| GYNAECOLOGY | | | | | | | | | | | | |
| Yamada | 1997 | NA | Cefmetazole 2g added to 0.5L of heparinized physiological saline | NR | NR | NR | NR | NA | NA | NA | AB: 11/22 (50) vs No-AB: 11/11 (100) | |
| Waters | 2000 | LeukoGuard RS (Pall Biomedical Products Co.) | No | 3.0 [0.6–7.7]† | 14/14 (100) | 1.3 [0.4–6.1]† | 13/14 (93) | 0.1 [0.0–0.9]† | 7/14 (50) | NA | NA | |
| Teare | 2015 | RS1 VAE (Pall Biomedical Products Co.) | No | 8 [1–84]§, total count 3,400 (1,278–52,200) | NA | 2 [1–25]§, total count 303 (188–1,245) | NA | 3 [1–14]§, total count 438 (98–2,115) | NA | NA | NA | |

(Continued)

| First author | Year | Type of LDF | Additional decontamination | Pre-wash contamination, quantified: | Pre-wash samples contaminated (%) | Post-wash/pre-LDF contamination, quantified: | Post-wash samples contaminated (%) | Post-LDF contamination, quantified: | Post-LDF samples contaminated (%) | Post-other decontamination, quantified: | Post-other decontamination samples contaminated (%): | Notes |
|---|---|---|---|---|---|---|---|---|---|---|---|---|
| EAR, NOSE AND THROAT SURGERY & ORAL AND MAXILLOFACIAL SURGERY | | | | | | | | | | | | |
| Lenzen | 2006 | WBF3 (Pall Biomedical Products Co.) | 4.5g/L piperacillin/tazobactam + 240 mg/L gentamicin, 1600 mg/L fluconazole with 2h incubation | $2.8 \times 10^4 \pm 3.3 \times 10^4$*, with a maximum of $10^5$ CFU/ml; $1.0 \pm 0.6$ species | NR | $2.6 \times 10^4 \pm 3.3 \times 10^4$*, with a maximum of $10^5$ CFU/ml; $1.6 \pm 0.7$ species | NR | $6.4 \times 10^3 \pm 2.3 \times 10^4$*, with a maximum of $10^5$ CFU/ml; $0.4 \pm 0.7$ species | NR | No longer detected (detection limit $10^2$ CFU/ml) | NR | |
| Wasl | 2016 | Not specified | No | NR | NR | NR | NR | NR | 0/8 | NA | NA | |
| EX-VIVO STUDIES | | | | | | | | | | | | |
| Marks | 1988 | NA | Positively charged antibacterial membrane (ABM) (CUNO, Microfiltration Products Division) | NR | NR | NR | NR | NR | NR | NR | NR | Bacterial removal of ABM[††] Aerobes: $99.28 \pm 0.29$ Anaerobes: $84.95 \pm 2.5$ Bacterial removal of cell washer[††] Aerobes: $87.47 \pm 5.36$ Anaerobes: $71.15 \pm 6.06$ |
| Waters | 2003 | LeukoGuard RS (Pall Biomedical Products Co.) | No | E. coli: 1,920 ± 452, S. aureus: 3,691 ± 5,152, P. aeruginosa: 1,970 ± 1,020, B. fragilis: 4,603 ± 1,480* | NR | E. coli: 440 ± 113; S. aureus: 436 ± 256; P. aeruginosa: 227 ± 113; B. fragilis: 1,039 ± 236* | NR | E. coli: 19 ± 16; S. aureus: 4 ± 7; P. aeruginosa: 0.6 ± 2; B. fragilis: 111 ± 74* | NR | NA | NA | %reduction (pre-wash to post LDF): E. coli: 99.0, S. aureus: 99.9, P. aeruginosa: 100, B. fragilis: 97.6 |
| Yost | 2017 | LeukoGuard RS (Pall Biomedical Products Co.) | No | 256,075 [4,300–504, 000]† | NA | NR | NA | 222,800 [1,200–424, 000]† | NR | NA | NA | %reduction: [-414.3 – 72.1]‡ |
| | | LeukoGuard BC2 Cardioplegia (Pall Biomedical Products Co) | No | [126–7,300] ** | NA | NR | NA | [91–200] ** | NR | NA | NA | %reduction: [27.8 – 97.6]‡ |

*(Continued)*

| First author | Year | Type of LDF | Additional decontamination | Pre-wash contamination, quantified: | Pre-wash samples contaminated (%) | Post-wash/pre-LDF contamination, quantified: | Post-wash samples contaminated (%) | Post-LDF contamination, quantified: | Post-LDF samples contaminated (%) | Post-other decontamination, quantified: | Post-other decontamination samples contaminated (%): | Notes |
|---|---|---|---|---|---|---|---|---|---|---|---|---|
| Hinson | 2020 | RS 40 µm leukocyte reduction filter (Haemonetics); 2 consecutive filters | No | E. coli: 1,128 ± 634.52, S. pseudintermedius: 773.33 ± 210.09, P. aeruginosa: 1,100.67 ± 529.51* | NR | E. coli = 178 ± 278.95, S. pseudintermedius = 64 ± 67.49, P. aeruginosa = 64.67 ± 110.76*; P < .0001 | NR | Post-LDF1: E. coli = 0.03 ± 0.1, S. pseudintermedius = 0 ± 0, P. aeruginosa = 0 + 0*; Post-LDF2: no contamination | NR | NA | NA | %reduction (pre-wash to post-LDF): E. coli: 99.9, S. pseudintermedius: 100, P. aeruginosa: 100 |

*Note.* AB: antibiotics, ACD: Acid Citrate Dextrose, BDA: Bile Duct Anastomosis, CFU: Colony Forming Unit, CG: Control Group, g: gram, L: litre, LDF: Leukocyte Depletion Filter, mg: milligram, NA: Not Applicable, NR: Not Reported, SD: Standard Deviation, SEM: Standard Error of the Mean, SG: Study Group, ug: microgram.

* mean CFU/mL ± SD.

† mean CFU/mL + range.

§ median + range.

‡ range.

** range CFU/mL.

†† % + SEM.

**Table 3. Clinical outcomes following reinfusion of salvaged blood.**

| First author | Year | Antibiotic prophylaxis | N transfused (ics group) | Bacteraemia | Total infectious complications | Description | Infection related to ics | Follow up duration | Other clinical outcomes |
|---|---|---|---|---|---|---|---|---|---|
| CARDIOVASCULAR SURGERY | | | | | | | | | |
| Davies | 1987 | Cloxacillin 1g at induction, every 6h for 24h | 25 (25) | None | 1/25 (4.0) | 1 respiratory infection with multisystem failure | No association | NR | NR |
| Ezzedine | 1991 | Cefazolin 2g at induction, every 6h for 48h | 401 (401) | 20/401 (5.0) | 24/401 (6.0) | 20 bacteraemia, 11 POWI, 4 osteitis, 5 mediastinitis (in 24 patients) | In 1 case cultures from salvaged blood matched pus cultures and post-operative cultures | Up to 3 months | NR |
| Bland | 1992 | Not stated | 38 (38) | 1/28 (3.6) | 1/28 (3.6) | One case of bacterae-mia, no other infectious complications reported. | No association | NR | NR |
| Reents | 1999 | Cefuroxime peri-op (not specified) | 10 (10) | None | None | NA | NA | NR | NR |
| Shindo | 2004 | Broad-spectrum pre-op and 72h post-op (not specified) | 52 (52) | None | None | NA | NA | 1 week | Cultures positive in 15/73 (21%) of thoracic operations vs 30/43 (70%) abdominal operation (P < 0.001) |
| Ishida | 2006 | Cefazolin 1g at induction, 2g during CPB, continued until POD5 | 35 (35) | None | None | NA | NA | NR | No correlation between the operation time and the count of bacteria |
| Schmidt | 2009 | Antibiotics at induction according to hospital protocol (not specified) | 6 (6) | NR | NR | NR | NR | NR | NR |
| Zhou | 2023 | Peri-op and intra-op prophylaxis (not specified) | 204 (204) | Culture+: 12/100 (12) Culture−: 4/104 (3.8) P=0.03 | Culture+: 22 (22) Culture−: 10 (9.6) | Culture+: 12 bacter-aemia, 6 pneumonia, 3 POWI, 1 UTI Culture−: 4 bacterae-mia, 2 pneumonia, 2 POWI, 2 UTI | ICS blood culture+ was an independent predictor (OR 2.62, 95% CI 1.16–5.90, P=0.02). Not reported whether isolated pathogen causing infection matched the organ-ism cultured from the salvaged blood. | 3 days post-operative cul-ture; 30-day mortality recorded | ICU stay (3.5 vs. 2 days, P<0.01); venti-lation time (20.45 vs. 13h, P=0.02); Inde-pendent predictors of ICS blood culture positivity: BMI ≥25 (OR 2.08), smoking (OR 1.99), surgery ≥277.5min (OR 5.58), case order 2 (OR 2.07), >9 OR staff (OR 1.19), all P ≤0.04 |

*(Continued)*

**Table 3.** (Continued)

| First author | Year | Antibiotic prophylaxis | N transfused (ics group) | Bacteraemia | Total infectious complications | Description | Infection related to ics | Follow up duration | Other clinical outcomes |
|---|---|---|---|---|---|---|---|---|---|
| **LIVER SURGERY** | | | | | | | | | |
| Kang | 1991 | Ampicillin 1g and cefotaxime 1g every 6h peri-op | 14 (14) | None | 1/14 | 1 UTI | No | 1 week | NR |
| Feltracco | 2007 | Standard broad-spectrum peri-op antibiotic therapy (not specified) | 38 (38) | None | None | NA | NA | Not explicitly stated, at least until POD3 | NR |
| Liang | 2010 | Sulperazon 2g/12 h or Tazocin 4.5g/12 h | 25 (12) | ICS: 2/12 (16.7%) C*: 5/13 (38.5%) | ICS: 4/12 (33.3%) C*: 8/13 (61.5%) | ICS: 3 respiratory, 1 UTI, 3 abdominal, 2 bacteraemia (in 4 patients) C*: 4 respiratory, 1 UTI, 2 POWI, 5 abdominal and 5 bacteraemia (in 8 patients) | No; none of the 12 patients that received salvaged blood received contaminated RBCs. | Until discharge | Pre-operative SBP and longer operative times independent risk factors for bacterial contamination (SBP: $p < 0.05$, OR = 20.1; operative time: $p < 0.01$, OR = 8.3) |
| Kim | 2022 | Cefotaxime 2g every 8h + ampicillin/sulbactam 3g every 6h from pre-op until POD3 | 29 (29) | None | None | None | NA | Not explicitly stated, at least until POD5 | No significant difference in contamination of ICS blood before versus after BDA: 44.8% vs. 31.0%; $p = 0.228$ |
| Kim | 2024 | Ampicillin/sulbactam 3g and cefotaxime 2g pre-op, continued or adjusted as needed | 30 (30) | Culture+: 3/10 (30) Culture-: 1/20 (5) (OR: 8.14, P=0.06) | 4/30 (13.3%) | 4/30 bacteraemia, no other infectious complications reported. | Probable in 3/4 bacteraemia cases; strain matched autologous blood | Not explicitly stated, at least until 2 weeks postoperative | No significant difference in contamination of ICS blood before versus after graft reperfusion: 23.3% vs. 36.7%; $p = 0.25$ |
| **ORTHOPAEDIC SURGERY** | | | | | | | | | |
| Wollinsky | 1997 | Yes (group B); cefuroxime 1.5g IV after induction | 40 (40) | None | 6/40 (10%) | No wound/pulmonary infections; no-AB group: 2 UTIs, AB-group 4 UTIs. | NR | Until discharge | |
| Nosanchuk | 2001 | Oxacillin 1g + gentamicin 80–100 mg pre-op, then 1g oxacillin every 4h intraoperatively | 18 (18) | None | None | NA | NA | NR | |

*(Continued)*

**Table 3.** (Continued)

| First author | Year | Antibiotic prophylaxis | N transfused (ics group) | Bacteraemia | Total infectious complications | Description | Infection related to ics | Follow up duration | Other clinical outcomes |
|---|---|---|---|---|---|---|---|---|---|
| Perez-Ferrer | 2016 | Cefazolin 35mg/kg after induction | 24 (24) | 1/24 (4.2) | 2/24 (8.3) | 2/24 (8.3%) had fever | One patient had positive culture of ICS blood with CNS | NR | |
| Perez-Ferrer | 2017 | Cefazolin 35mg/kg after induction | 20 (20) | None | None | NA | NA | NR | |
| Kruger | 2024 | Reimplantation group: tailored to prior PJI pathogen, control: cefazolin/clindamycin | 20 (20) | None | None | NA | NA | Intermediate follow up during inpatient stay: mean 37.6±13.2 days (range: 18–73). Long term follow up: mean 19.2 months (range 11–39) | |
| **TRAUMA SURGERY** | | | | | | | | | |
| Timberlake | 1988 | Broad-spectrum cephalosporins peri-op (not specified) | 11 (11) | 1/11 (9.1) | 3/11 (27.3) | 2 POWI, 1 bacteraemia + necrotizing fasciitis (nosocomial) | 2 patients had superficial POWI with the same organisms cultured from ICS blood. | Until discharge | |
| Ozmen | 1992 | Broad-spectrum antibiotics pre-op, continued until POD1, 3 or 5 (not specified) | 70 (20) | ICS: 3/20 (15) C*: 2/50 (4) | ICS: 13/20 (65%) C*: 12/50 (24%) | ICS: 3 bacteraemia (15%), 4 pulmonary (20%), 2 UTI (10%), 4 abscesses (20%) C*: 2 bacteraemia (4%), 5 pulmonary (10%), 5 abscesses (10%); no significant differences between groups when stratified for injury severity | No association between organisms causing UTI, bacteraemia, or pulmonary infections and those cultured from ICS blood. POWIs and abscesses caused by enteric flora, predominantly E. coli | NR | Mortality ICS: 4/20 (20%); 2 patients died within 72 hours of admission from shock and coagulopathy. 2 patients died >2 weeks after admission of sepsis related multi organ failure. Mortality C*: 0/50 |
| Bowley | 2006 | Prophylactic antibiotics according to surgeons schedule (not specified) | 44 (21) | ICS: 5/13 (38.5) C*: 7/13 (53.8) (P=0.69) | ICS: 5/13 (38.5) C*: 7/13 (53.8) | Sepsis in ICS: 5/13 (38.5%) Sepsis in C*: 7/13 (53.8%) (P=0.69) No other infectious complications reported. | No association between initial microbiology of the reinfused blood and subsequent infective episodes. | Until discharge | Survival ICS vs C*: 7/21 (33.3%) vs 8/23 (35%) (P=NS). Survival patients with enteric injury ICS vs C*: 7/18 (38.8%) vs (23.5%) (P=0.47). No difference in duration of hospital stay between ICS and C* group |

*(Continued)*

**Table 3.** (Continued)

| First author | Year | Antibiotic prophylaxis | N transfused (ics group) | Bacteraemia | Total infectious complications | Description | Infection related to ics | Follow up duration | Other clinical outcomes |
|---|---|---|---|---|---|---|---|---|---|
| **GYNAECOLOGY** | | | | | | | | | |
| Yamada | 1997 | Cefazolin, flomoxef, cefapirin, cefminox, or cefmetazole (2x/day post-op) (duration not specified) | 40 (40) | NR | 6/40 | 6/40 (15%) had mild fever <39°C; 1–2 day duration; 0 major adverse events | NR | NR | |
| **EAR, NOSE AND THROAT SURGERY & ORAL AND MAXILLOFACIAL SURGERY** | | | | | | | | | |
| Locher | 1992 | Clindamycin (n = 13) or cotrimoxazole (n = 5) (dosage and duration not specified) | 36 (18) | 15min post-transfusion: 15/18; 2h: 1/18; 24h: 0/18 | ICS: 5/18 (27.8%) C*: 1/18 (12.5%) | ICS: 5/18 (27.8%) fever >39°C C*: 1/18 (12.5%) fever >39°C | NR | Not explicitly stated, at least 24h | |
| Wasl | 2016 | Intra-op and 48h post-op broad-spectrum antibiotics (not specified) | 10 (10) | LDF: 0/8 No LDF: 2/8 | 2/10 (20%) | Transient bacteraemia in 2 patients (no LDF used) | Bacteraemia was caused by same organism as cultured from ICS blood | ≥18 months | |
| **OTHER/MIXED** | | | | | | | | | |
| Sugai | 2001 | NR | 287 (30) | None | None | NA | NA | NR | |
| Kudo | 2004 | Flomoxef 2g until POD5 | 37 (37) | None | None | NA | NA | 6 months – 4 years and 10 months | |

*Note.* AB: Antibiotics, BMI: Body Mass Index, C: Control group, CNS: Coagulase-Negative Staphylococci, CPB: Cardiopulmonary Bypass, ENT: Ear, Nose, and Throat, ICS: Intraoperative Cell Salvage, ICU: Intensive Care Unit, IV: Intravenous, LDF: Leukocyte-Depletion Filter, NA: Not Applicable, NR: Not Reported, OR: Odds Ratio, PJI: Periprosthetic Joint Infection, POD: Postoperative Day, POWI: Postoperative Wound Infection, RBC: Red Blood Cell, SBP: Spontaneous Bacterial Peritonitis, UTI: Urinary Tract Infection.

*allogeneic transfusion only.

**Table 4. Quality assessment of cohort studies included.**

| First author | Year | Selection | Comparability | Outcome | Overall quality |
|---|---|---|---|---|---|
| Andrews | 1983 | ★★★ | | ★★★ | POOR |
| Bland | 1992 | ★★★ | | ★★★ | POOR |
| Ezzedine | 1991 | ★★★ | | ★★★ | POOR |
| Feltracco | 2007 | ★★★ | | ★★★ | POOR |
| Gigengack | 2021 | ★★★ | | ★★ | POOR |
| Ishida | 2006 | ★★★ | | ★★ | POOR |
| Jeng | 1998 | ★★★ | | ★ | POOR |
| Kang | 1991 | ★★★ | | ★★★ | POOR |
| Khan | 1975 | ★★★ | | ★★★ | POOR |
| Kim | 2022 | ★★★ | | ★★★ | POOR |
| Kim | 2024 | ★★★★ | ★★ | ★★★ | GOOD |
| Kruger | 2024 | ★★★★ | ★★ | ★★★ | GOOD |
| Kudo | 2004 | ★★★ | | ★★★ | POOR |
| Lenzen | 2006 | ★★★ | | ★★★ | POOR |
| Liang | 2010 | ★★★ | | ★★★ | POOR |
| Locher | 1992 | ★★★ | | ★★★ | POOR |
| Luque-Oliveros | 2020 | ★★ | | ★ | POOR |
| Marks | 1988 | ★★★ | | ★★★ | POOR |
| Nosanchuk | 2001 | ★★★ | | ★★★ | POOR |
| Ozmen | 1992 | ★★★ | | ★★★ | POOR |
| Perez-Ferrer | 2016 | ★★★ | | ★★★ | POOR |
| Reents | 1999 | ★★★ | | ★★★ | POOR |
| Schmidt | 2009 | ★★★ | | ★★★ | POOR |
| Shindo | 2004 | ★★★ | | ★★★ | POOR |
| Sugai | 2001 | | | ★★★ | POOR |
| Taere | 2015 | ★★★ | | ★★ | POOR |
| Timberlake | 1988 | ★★★ | | ★★★ | POOR |
| Wasl | 2016 | ★★★ | | ★★★ | POOR |
| Waters | 2000 | ★★★ | | ★★★ | POOR |
| Yamada | 1997 | ★★★ | | ★★★ | POOR |
| Zhou | 2023 | ★★★ | ★★ | ★★★ | GOOD |

## Cardiovascular surgery

Eleven studies reported on the use of ICS during cardiothoracic or vascular procedures [24–34].

**Bacterial contamination.** All studies reported bacterial contamination of salvaged blood, with positive culture rates ranging from 12.7% to 96.8% [24–34].

**Reduction strategies.** One study routinely added antibiotics to the anticoagulant solution as part of the ICS protocol, without a comparator group. Bacterial contamination persisted in 12.7% of processed samples [32]. No other studies applied additional decontamination strategies.

**Clinical outcomes.** Eight studies reinfused salvaged blood [25–28,30–33]. One study did not report clinical outcomes [31]. In four studies no significant infectious complications attributable to ICS were observed [26–28,33]. One study reported a single case of postoperative bacteraemia without clinical signs of sepsis. The organism isolated from the patient was not detected in the salvaged blood sample [25]. Another study reported infectious complications in 6% of patients (n=20), but

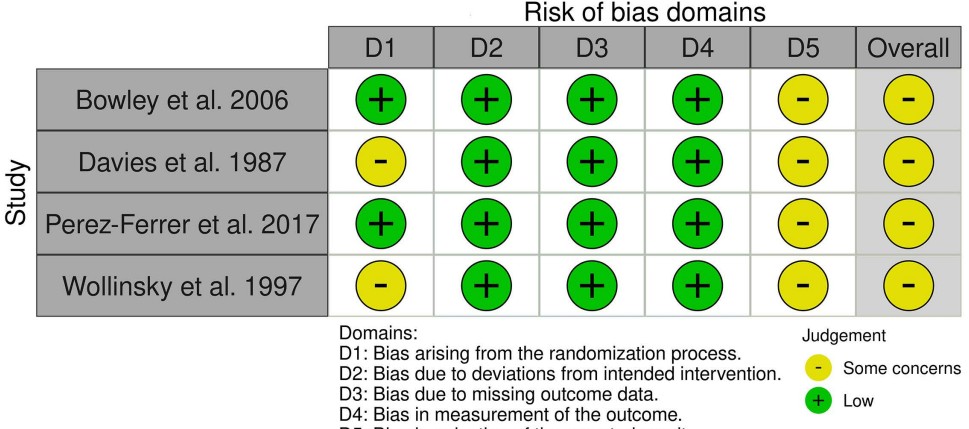

Risk of bias domains

| Study | D1 | D2 | D3 | D4 | D5 | Overall |
|---|---|---|---|---|---|---|
| Bowley et al. 2006 | + | + | + | + | - | - |
| Davies et al. 1987 | - | + | + | + | - | - |
| Perez-Ferrer et al. 2017 | + | + | + | + | - | - |
| Wollinsky et al. 1997 | - | + | + | + | - | - |

Domains:
D1: Bias arising from the randomization process.
D2: Bias due to deviations from intended intervention.
D3: Bias due to missing outcome data.
D4: Bias in measurement of the outcome.
D5: Bias in selection of the reported result.

Judgement
- Some concerns
+ Low

**Fig 2. Quality assessment of RCTs included.**

only in one patient this could be directly linked to the bacteria cultured from the ICS blood sample [32]. Another study found a significantly higher postoperative infection rate in patients who received culture-positive salvaged blood compared to those who did not (22.0% vs. 9.6%, p=0.02), identifying bacterial contamination as an independent risk factor for infection [30]. The study does not report whether the organisms isolated from patients matched those cultured from the salvaged blood.

## Liver surgery

Five studies reported on ICS during liver transplantation [16,35–38] and one during hemihepatectomy [31]. Liver surgery is particularly prone to contamination due to ascitic fluid, bile spillage, and bacterial translocation from the intestine, especially in patients with end-stage liver disease.

**Bacterial contamination.** Bacterial contamination of salvaged blood was consistently reported across all six studies [16,31,35–38]. In one study, no contamination was detected in the final processed blood after leukocyte filtration while earlier processing stages did demonstrate positive cultures [16]. The contamination rates of the final ICS blood ranged from 0% to 68.4%.

**Reduction strategies.** Three studies assessed the efficacy of LDFs [16,37,38], reporting reductions in contamination rates from 36.4% to 100%. Two studies demonstrated statistically significant conversion rates [16,37]. Residual contamination varied between 0% to 23.3%.

**Clinical outcomes.** Salvaged blood was reinfused in five studies [16,35–38]. Three reported no cases of postoperative bacteraemia [16,35,36], while two other studies reported bacteraemia in up to 16.7% of patients [37,38]. In one of these studies, none of the transfused patients received culture-positive ICS blood [37]. In the other, 75% of bacteraemia cases involved organisms that matched those cultured from the salvaged blood [38].

## Orthopaedic surgery

Five studies addressed the use of ICS in orthopaedic procedures [39–43]

**Bacterial contamination.** Bacterial contamination of salvaged blood was consistently reported, with contamination rates ranging from 15 to 75% [39–43].

**Reduction strategies.** One study demonstrated that the addition of vancomycin to the wash solution completely eliminated bacterial contamination in all samples, compared to a 50% contamination rate in the control group (p=0.016) [42]. No additional bacterial reduction techniques were employed in the other studies.

**Clinical outcomes.** All five studies reinfused salvaged blood [39–43]. One study described a single case of bacteraemia, with the isolated organism matching that cultured from the ICS blood sample [41]. None of the studies reported postoperative sepsis or clinically significant infections attributable to the reinfusion of ICS blood.

### Trauma surgery

Three studies reported on ICS in cases of penetrating abdominal trauma [44–46]. In this setting, contamination of salvaged blood with enteric contents is a major concern due to frequent involvement of hollow viscus injuries.

**Bacterial contamination.** Bacterial contamination of salvaged blood was consistently reported in all studies [44–46]. Contamination varied, with one study reporting 16% of units of ICS blood being contaminated, while another study reported contamination in 100% of patients.

**Reduction strategies.** No additional bacterial reduction techniques were applied

**Clinical outcomes.** In all three studies ICS blood was reinfused [44–46]. No consistent association was found between positive cultures of reinfused blood and post-operative bacteraemia. Mortality in these studies was primarily attributed to injury severity and exsanguination.

### Gynaecology

Three studies reported on the use of ICS in gynaecological procedures [15,47,48], which are prone to contamination due to proximity to vaginal or gastrointestinal flora.

**Bacterial contamination.** Bacterial contamination of ICS blood was observed in all studies. In abdominal hysterectomy for uterine myoma, 66.7% of processed blood samples were culture-positive [47]. During caesarean section, all shed blood samples were culture-positive. After washing, 93% remained culture-positive, and 50% of samples were still positive after final processing [15]. In vaginal delivery cases, contamination was present in all samples, with median bacterial concentrations of 8 colony-forming units per millilitre (CFU/mL) (range: 1–84) before processing and 2 CFU/mL (range: 1–25) after washing [48].

**Reduction strategies.** LDFs were used in two studies [15,48], with variable reductions in bacterial contamination reported. One study additionally investigated the use of antibiotics added to the heparinized saline solution during processing, resulting in a 50% reduction in the amount of culture-positive ICS blood [47].

**Clinical outcomes.** Only one study reinfused ICS blood and no significant postoperative infections or adverse effects related to reinfusion were observed [47].

### Ear, nose and throat (ENT) and oral and maxillofacial (OMS) surgery

Three studies reported on the use of ICS in ENT and OMS surgical procedures [49–51]. These procedures typically involved contaminated surgical fields due to proximity to the oral or nasal cavity, which are colonized by commensal bacteria.

**Bacterial contamination.** Bacterial contamination of salvaged blood was observed in all studies. One study reported complete bacterial clearance in the final product, but contamination was detected in earlier processing stages [50]. Final contamination rates of ICS blood ranged from 0% to 100%.

**Reduction strategies.** Two studies examined the use of LDFs [50,51]. In one study, no bacterial contamination was detected after leukoreduction [51]. The other study reported that LDF alone led to partial bacterial reduction, but complete elimination was achieved through a multistep protocol that included antimicrobial treatment of the salvaged red blood cells [50].

**Clinical outcomes.** In two studies, salvaged blood was reinfused [49,51]. No major infectious complications were reported. Transient bacteraemia, caused by bacteria also cultured from ICS blood, was observed in two patients where no LDF was used [51].

### Other

Two studies evaluated ICS in burn excisional surgery [52,53], one assessed ICS in neurosurgical procedures [54], and one investigated contamination across different types of autologous transfusion without specifying the type of surgery [55].

**Bacterial contamination.** In burn excisional surgery, bacterial contamination was detected in all ICS blood samples [52,53]. During neurosurgical procedures, contamination was observed in 46.7% of samples [54]. In the study comparing autologous transfusion methods, contamination was reported in 33.3% of ICS samples, compared to 4.8% in samples from preoperatively donated autologous blood and 2.6% in haemodilution samples [55].

**Reduction strategies.** No additional reduction strategies were employed

**Clinical outcomes.** Two studies reinfused ICS blood and reported no adverse events following reinfusion [54,55].

### Ex-vivo studies

Three ex-vivo experimental studies investigated the efficacy of ICS systems combined with LDFs for bacterial decontamination of blood products [56–58]. One study investigated the effect of washing and centrifugation on contamination alone [59]. One study reported on the efficacy of an antibacterial membrane to remove fecal contamination from a RBC and plasma suspension and compared this to the efficacy of a cell washer [60].

## Discussion

This scoping review studied the existing evidence regarding the use of ICS in bacterially contaminated surgical fields. A total of 39 studies were included, encompassing 1654 patients who underwent ICS. Overall, the findings indicate that bacterial contamination of salvaged blood is common across a wide range of surgical contexts, including those not typically classified as contaminated, such as cardiovascular or orthopaedic surgery. Most common contaminants were skin commensals. Other contaminants included organisms typically originating from the oral, gastrointestinal, or urogenital tract, depending on the anatomical site and type of procedure performed. In some cases, environmental bacteria were also identified, suggesting possible contamination from surgical instruments, suction systems or the operating room environment.

Several studies evaluated techniques aimed at reducing bacterial contamination during ICS, primarily the use of LDFs (10 studies) and, the addition of antibiotics to the anticoagulant or wash solution (4 studies). LDFs were found to effectively reduce bacterial contamination in several studies, in some cases even to undetectable levels. However, the extent of bacterial reduction varied widely. Similarly, the addition of antibiotics to the cell salvage process reduced the number of culture-positive samples, suggesting its potential as a bacterial reduction strategy. While current evidence does not allow for a definitive recommendation of any single decontamination strategy, the routine use of LDFs appears justifiable in cases where bacterial contamination is suspected or confirmed.

Salvaged blood was reinfused in 26 of the 39 included studies, involving a total of 1203 patients. Postoperative infectious complications of any kind were reported in 15 of these studies. A total of 55 cases of postoperative bacteraemia were documented across these studies. However, only five studies described a plausible microbiological link between the reinfused blood and subsequent infection, identifying a total of nine patients in whom the causative pathogen matched the organism cultured from the salvaged blood. In cardiovascular surgery, only one study identified bacterial contamination of ICS blood as an independent predictor for postoperative infection [30]. However, this study did not report whether the isolated pathogen matched the organism cultured from the salvaged blood. On the other hand, it has to be mentioned that the studies included in this review lacked the statistical power to properly detect an increased risk of infection.

These findings are in line with previous studies suggesting that the presence of bacteria in ICS blood does not necessarily translate into postoperative infection [25,36,43]. These observations support the hypothesis that low levels of bacterial contamination, especially by low-virulence organisms, may be cleared by the immune system without clinical

consequences [61,62]. Other clinical fields have similar methodological challenges, where advanced statistical models have been applied to physiological signals to identify subtle but clinically relevant features [63]. This might be a solution to deal with the interaction of the immune system and reinfusion of bacterially contaminated blood. Although, in some contexts such as cardiac surgery, ICS was associated with higher postoperative infection rates, this effect was counterbalanced by the reduced exposure to allogeneic transfusion. As a result, the overall infection risk did not increase [64].

Furthermore, bacterial load alone does not appear to be a reliable predictor of clinical outcome. Previous research has shown that a higher bacterial load does not always translate to symptomatic infection. In a study where eight patients received bacterially contaminated platelets, five patients remained asymptomatic despite bacterial concentrations ranging from $10^2$ to $10^{11}$ CFU/mL. In contrast, symptomatic cases had bacterial loads ranging from $10^6$ to $10^8$ CFU/mL [65].

In addition, appropriate perioperative antibiotic prophylaxis may further reduce the risk of infection following reinfusion of bacterially contaminated salvaged blood. Antibiotics that are routinely administered during surgery to prevent surgical site infections may also provide effective coverage against potential contaminants introduced through ICS [39,66].

Lastly, intraoperative environmental and procedural factors appear to play a crucial role in the bacterial contamination of salvaged blood. One study reported that a longer operative duration, a greater number of OR staff present, and a higher surgical case order were all independent risk factors for positive bacterial cultures in salvaged blood [30]. These factors are also well-established risk factors for surgical site infections [67]. This overlap suggests that factors known to increase the risk of surgical site infections, such as extended operative time, frequent door openings, increased OR room traffic, and potentially other perioperative variables like inspired oxygen concentration ($FiO_2$), may likewise contribute to the risk of bacterial contamination in salvaged autologous blood.

## Strengths and limitations of the evidence

A key strength of this scoping review is the inclusion of studies from a wide range of surgical specialties, providing a comprehensive overview of ICS in the setting of bacterial contamination. To our knowledge, this is the first review to systematically study the evidence on ICS use in this context. Another methodological strength is the unrestricted time frame applied during study selection. This approach was particularly important because, although advances in surgical technique and sterility may have reduced contamination rates over time, the clinical consequences of reinfusing contaminated blood remain fundamentally unchanged. Excluding older studies solely based on publication date would therefore risk disregarding valuable data that remains relevant to the objectives of this review. Furthermore, the consistency in reported contamination rates, alongside the generally low incidence of adverse clinical outcomes, adds to the credibility of the findings. Nonetheless, several important limitations must also be acknowledged.

Most studies were small, single-centred, and observational. There was substantial heterogeneity in study design, ICS protocols and devices used, bacterial detection methods, and clinical outcome definitions. Differences in bacterial detection methods included different culture media, volumes, incubation times and reporting of outcomes (e.g., only positivity, or also bacterial counts). Variations in patient characteristics, use of antimicrobial prophylaxis and perioperative protocols across institutions further complicate the interpretation and generalisability of the findings. Furthermore, several studies used outdated ICS devices, which may have inferior processing capabilities compared to modern systems. Many of the included studies were conducted decades ago, when intraoperative sterility standards differed from current practice. Improvements such as stricter infection control protocols and refined surgical techniques have reduced (environmental) contamination risk considerably [68–70]. As a result, direct comparison of older and more recent study outcomes is challenging. However, despite advancements in technology and surgical protocols, more recent studies do not consistently report lower contamination rates. This suggests that factors beyond equipment generation and surgical era may continue to contribute to ICS-related contamination.

This scoping review itself has limitations. The methodological quality or risk of bias was conducted for the studies included, except for the ex-vivo studies. However, due to the heterogeneity regarding population, design and outcome

measurements, the methodological quality of the studies was not synthesized. The methodological quality is only presented to contextualize the findings. Secondly, the scope and heterogeneity of the topic precluded quantitative synthesis or meta-analysis, which limits the ability to compare findings across settings or interventions in a standardized way. However, due to the broad scope and methodological heterogeneity of the current evidence base, a systematic review was not considered appropriate.

## Implications for clinical practice

Given the current body of evidence, ICS may be cautiously considered in selected cases, particularly where expected blood loss is high or allogeneic donor blood is not widely available. The consistent finding that reinfusion of bacterially contaminated blood is not directly associated with postoperative infection suggests that ICS may be feasible under appropriate conditions. Moreover, ICS may offer protective benefits by avoiding allogeneic transfusion and its associated immunomodulatory effects.

As earlier mentioned manufacturers of ICS devices list active infection or gross contamination as contraindications due to the risk of bacteraemia and the UK Cell Salvage Action Group, generally discourage ICS in infected settings [11–13]. We believe these recommendations remain robust. However, this review highlights that bacterial contamination is present in specialties generally not considered as contaminated, like cardiovascular and orthopaedic surgery. Therefore, each clinical decision should be guided by a case-by-case assessment that considers contamination severity, contamination source, patient immune status, and the availability of decontamination techniques such as leukocyte depletion filters and antibiotic prophylaxis. In situations with gross contamination, such as visible enteric contents, reinfusion may carry a higher risk and should be approached with particular caution. Conversely, in cases of minimal contamination involving low-virulence organisms such as skin flora the risk may be more acceptable, especially when decontamination techniques are applied.

Use of ICS in contaminated settings should ideally occur within standardized protocols or clinical trials to ensure patient safety and contribute to the evidence base. Until higher-quality data become available, clinical discretion remains key.

## Implications for future research

This review highlights several critical knowledge gaps that should guide future investigations. Firstly, the identification of safe contamination thresholds is needed. Future studies should aim to correlate quantitative bacterial loads with clinical outcomes, to determine whether there are tolerable levels of contamination below which reinfusion is safe. Importantly, such assessments should also account for the type and virulence of the contaminating organisms, as some bacteria are more likely to cause clinical infection than others.

Secondly, the effectiveness of specific decontamination techniques, such LDFs or the addition of antibiotics, requires more rigorous evaluation. Comparative trials could help identify which methods are most effective in reducing bacterial contamination.

Thirdly, prospective, large-scale clinical studies are needed to determine safety of ICS in contaminated fields. Ideally, these would include multicentre cohort studies or controlled trials comparing ICS versus no ICS (or versus allogeneic transfusion) across contaminated surgical scenarios. These studies should systematically document intraoperative factors that may influence contamination risk, such as case order, duration of surgery, frequency of door openings, and the number of personnel present in the operating room. In addition to infection rates, broader clinical outcomes, such as ICU and hospital length of stay, need for allogeneic transfusion, mortality, and cost-effectiveness, should also be assessed.

Establishing the determinants that influence the safety of ICS in contaminated surgical settings is critical. The influence of factors such as microbial resistance patterns, the degree of contamination, and patient-specific variables must be systematically evaluated to better understand when ICS can be safely used. These efforts will be essential for the development of context-specific, evidence-based guidelines for ICS in contaminated surgical fields.

## Conclusion

Bacterial contamination of salvaged blood was frequently observed across a wide range of surgical disciplines, including procedures not typically classified as contaminated. Decontamination strategies such as leukocyte depletion filtration and antibiotic additives were implemented in a subset of studies and demonstrated variable effectiveness in reducing bacterial contamination. Among 1203 patients who were reinfused salvaged blood, 55 cases of postoperative bacteraemia were identified. A potential microbiological link between salvaged blood and infection was described in only nine patients across five studies. These findings suggest that reinfusion of bacterially contaminated salvaged blood is not consistently associated with adverse clinical outcomes. However, substantial heterogeneity in methodologies and generally insufficient sample sized to detect rare adverse events limits the ability to draw definitive conclusions. Future research should aim to define safe contamination thresholds, evaluate the effectiveness of decontamination techniques, and assess clinical outcomes in standardized and controlled settings with sufficient sample size to detect adverse events.

## Supporting information

**S1 Table. (Species).**
(DOCX)

**S2 Table. (Pre- and post-wash).**
(DOCX)

**S3 Table. (Bacterial load).**
(DOCX)

**S4 Table. (Prisma checklist filled).**
(DOCX)

**S1 Appendix. Search strategies.**
(DOCX)

## Acknowledgments

The authors wish to thank dr. W. Bramer, Information Specialist, Erasmus MC Medical Library, for developing and updating the search strategy.

## Author contributions

**Conceptualization:** Joeri Slob, Pim Hondebrink, Ankie W.M.M. Koopman - van Gemert, Margriet E. van Baar, Cornelis H. van der Vlies, Seppe S.H.A. Koopman.

**Data curation:** Joeri Slob, Pim Hondebrink.

**Formal analysis:** Joeri Slob, Pim Hondebrink.

**Investigation:** Joeri Slob, Pim Hondebrink.

**Methodology:** Joeri Slob, Pim Hondebrink.

**Project administration:** Joeri Slob, Pim Hondebrink.

**Resources:** Joeri Slob, Pim Hondebrink.

**Software:** Joeri Slob, Pim Hondebrink.

**Supervision:** Ankie W.M.M. Koopman - van Gemert, Margriet E. van Baar, Cornelis H. van der Vlies, Seppe S.H.A. Koopman.

**Validation:** Joeri Slob, Pim Hondebrink.

**Visualization:** Joeri Slob, Pim Hondebrink.

**Writing – original draft:** Joeri Slob, Pim Hondebrink, Ankie W.M.M. Koopman - van Gemert, Margriet E. van Baar, Cornelis H. van der Vlies, Seppe S.H.A. Koopman.

**Writing – review & editing:** Joeri Slob, Pim Hondebrink, Ankie W.M.M. Koopman - van Gemert, Margriet E. van Baar, Cornelis H. van der Vlies, Seppe S.H.A. Koopman.

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
