## [Decision Letter · Decision Letter 0]

7 Oct 2025

Dear Dr. Koopman,

We look forward to receiving your revised manuscript.

Kind regards,

*
**Ali Amanati**
*

*
**Academic Editor**
*

*
**PLOS ONE**
*

Journal Requirements:

[This work is supported by the Dutch Burns Foundation (grant no. 19.102). The funder had no influence on the design or content of this manuscript.].

3. Please amend the manuscript submission data (via Edit Submission) to include authors J Slob, P Hondebrink, AWMM Koopman – van Gemert, ME van Baar, CH van der Vlies, and JSHA Koopman.

4. Please amend your authorship list in your manuscript file to include author Seppe Koopman, Joeri Slob, Pim Hondebrink, Ankie W.M.M. Koopman - van Gemert, Margriet E. van Baar, and Cornelis H. van der Vlies.

Additional Editor Comments:

Dear Authors,

Your manuscript [PONE-D-25-38734] has passed the review stage and is ready for revision. To ensure the editor and reviewers can recommend that your revised manuscript be accepted, please pay careful attention to each comment posted under this email. This approach will help us avoid future clarifications and revisions, allowing us to move swiftly to a decision.

Technical points:

‎1. Please provide a point-by-point response to the Editor and reviewer's comments

‎2. Please highlight all the amends on your manuscript with a yellow color

‎3. Use line numbering and page number in the next submission‎

Reviewers' comments:

Reviewer's Responses to Questions

**Comments to the Author**

1. Is the manuscript technically sound, and do the data support the conclusions?

Reviewer #1: Yes

Reviewer #2: Yes

Reviewer #3: Yes

2. Has the statistical analysis been performed appropriately and rigorously?

Reviewer #1: N/A

Reviewer #2: Yes

Reviewer #3: No

3. Have the authors made all data underlying the findings in their manuscript fully available?

Reviewer #1: Yes

Reviewer #2: Yes

Reviewer #3: Yes

4. Is the manuscript presented in an intelligible fashion and written in standard English?

Reviewer #1: Yes

Reviewer #2: Yes

Reviewer #3: Yes

Reviewer #1: This is the first scoping review mapping the use of intra-operative cell salvage (ICS) in bacterially contaminated surgical fields. The topic is highly relevant and the JBI/PRISMA-ScR methodology is generally followed. Nevertheless, the absence of any risk-of-bias or quality appraisal, together with imprecise definitions of “safe contamination threshold” and “causal infection”, limits the strength of the conclusions. The discussion insufficiently addresses the conflict between manufacturers’ contraindications and current guidelines, and the clinical recommendations may over-interpret the evidence.

1.Reviewers are required to assess methodological quality where feasible

2.Studies used different culture media, volumes, incubation time; some reported CFU/mL, others only positivity.

3.Conclusion states “reinfusion… is not consistently associated with adverse outcomes”, but many studies lacked power to detect infection difference.

Reviewer #2: Cell Salvage in Bacterially Contaminated Surgical Fields – a Scoping Review

I believe the summary could provide more definitive information on the subject, even though it remains controversial.

The introduction is lengthy, although very well-written, but could be adjusted to engage the reader.

The methods are very clear, with precise information. Additionally, the details are adequate, as are the inclusion and exclusion criteria.

The results are adequate, and the tables are very informative. The division of strategies by type of surgery seems quite appropriate.

The discussion and conclusion are well-founded and seem very appropriate to me.

I missed references and approaches to other surgeries, such as cesarean sections.

https://pubmed.ncbi.nlm.nih.gov/?term=+Intra-operative+cell+salvage

Reviewer #3: This is a well-conducted scoping review on intraoperative cell salvage (ICS) in bacterially contaminated surgical fields. The topic is highly relevant to clinical practice, and the authors followed established methodological frameworks (JBI, PRISMA-ScR). The manuscript is comprehensive and clearly structured, but some revisions would further strengthen its clarity, methodological rigor, and clinical impact.

While this is a scoping review rather than a systematic review, a brief assessment of study quality or risk of bias would help contextualize the findings. For example, highlighting differences between RCTs, prospective observational studies, and ex vivo studies could provide readers with a clearer sense of evidence strength.

In the Discussion section (around lines 343–369), where the authors discuss that “bacterial contamination does not necessarily lead to infection, possibly because low-level contamination can be cleared by the immune system,” they add the following:“Similar methodological challenges are seen in other clinical fields, where advanced statistical models have been applied to physiological signals to identify subtle but clinically relevant features (cite: Applied statistical methods for identifying features of heart rate that are associated with nicotine vaping).”

Please ensure consistent use of terms: “salvaged red blood cells (sRBCs)” vs. “ICS blood.”

Standardize abbreviations (e.g., ICS, LDF, CFU/mL) across text, tables, and figures.

Parts of the discussion (e.g., lines 343–369) contain repeated sentence structures such as “was found to” and “was reported to.” Rephrasing for variety would improve readability.

The conclusion could be more concise, emphasizing key clinical take-home messages for practitioners.

**Do you want your identity to be public for this peer review?** For information about this choice, including consent withdrawal, please see our Privacy Policy

Reviewer #1: **Yes: ** Caoyulong

Reviewer #2: **Yes: ** Vicente Sperb Antonello

Reviewer #3: No

---

## [Author Response · Author response to Decision Letter 1]

21 Nov 2025

Dear Editor,

We would like to thank the editor and reviewers for their comments and the opportunity to improve our manuscript. We have improved our manuscript according to the comments in a structured point-by-point approach. Below you can find the original comments from the reviewers as following:

italics

Our responses are presented in bold. Changes in the manuscript are made and marked by track-changes and highlighted in yellow in the track-changes version. The clean version of the manuscript is also presented.

Kind regards,

S.H.A. Koopman

Reviewer #1:

This is the first scoping review mapping the use of intra-operative cell salvage (ICS) in bacterially contaminated surgical fields. The topic is highly relevant and the JBI/PRISMA-ScR methodology is generally followed. Nevertheless, the absence of any risk-of-bias or quality appraisal, together with imprecise definitions of “safe contamination threshold” and “causal infection”, limits the strength of the conclusions. The discussion insufficiently addresses the conflict between manufacturers’ contraindications and current guidelines, and the clinical recommendations may over-interpret the evidence.

1.Reviewers are required to assess methodological quality where feasible

2.Studies used different culture media, volumes, incubation time; some reported CFU/mL, others only positivity.

3.Conclusion states “reinfusion… is not consistently associated with adverse outcomes”, but many studies lacked power to detect infection difference.

We would like to thank the reviewer for the critical appraisal.

Nevertheless, the absence of any risk-of-bias or quality appraisal, together with imprecise definitions of “safe contamination threshold” and “causal infection”, limits the strength of the conclusions.

1.Reviewers are required to assess methodological quality where feasible

Methodological quality assessment is not usual in scoping reviewers. However, as two of three reviewers mentioned assessing methodological quality we have performed a quality assessment using two tools. The Risk of Bias 2 tool for the Randomized controlled trials and the Newcastle-Ottawa Scale assessment for cohort studies included in this review. This was added to the methodology section and the results section of the manuscript.

“Methodological quality

The quality assessment of articles included was carried out by JS and PH independently. Whenever applicable a third reviewer (AK) arbitrated. The revised Cochrane risk of bias 2 tool for randomised controlled trials (RoB 2) was used for the included randomised controlled trials [22]. This tool uses stratification into five domains to detect potential bias. For included articles other than randomised controlled trials the Newcastle-Ottawa Score was used [23]. This tool uses stratification into three domains to score quality. Ex-vivo studies were not assessed for methodological quality.”

“Table 4 and Figure 2 summarize the quality assessments of RCTs (n=4) and cohort studies (n=31), except for the ex-vivo studies. The RCTs were assessed as some concerns (n=4). The cohort studies were assessed as poor (n=28) or good (n=3).”

A table 4 and figure 2 were added in the main paper below table 3.

Considering the safe contamination threshold and causal infection the following was added:

“Therefore, we would advise to use cell salvage in known bacterial contaminated areas on a case by case basis, after careful evaluation of the pros and cons, until future research defined safe contamination thresholds, evaluated the effectiveness of decontamination techniques, and assessed clinical outcomes in standardized and controlled settings.” was added in the abstract of the paper.

The discussion insufficiently addresses the conflict between manufacturers’ contraindications and current guidelines, and the clinical recommendations may over

interpret the evidence.

We agree that we have not described our clinical recommendations along with the manufacturers’ recommendations and guidelines. We have added sentences in the discussion to provide context alongside our recommendations.

“As earlier mentioned manufacturers of ICS devices list active infection or gross contamination as contraindications due to the risk of bacteraemia and the UK Cell Salvage Action Group, generally discourage ICS in infected settings (11-13). We believe these recommendations remain robust. However, this review highlights that bacterial contamination is present in specialties generally not considered as contaminated, like cardiovascular and orthopaedic surgery. Therefore…” was added.

2.Studies used different culture media, volumes, incubation time

some reported CFU/mL, others only positivity.

We agree that the studies included used different culture media, volumes, incubation time; whether or not highlighted by quantification like colony forming units. Although it was already mentioned in the discussion, we have added two lines to further emphasize this point.

“Differences in bacterial detection methods included different culture media, volumes, incubation times and reporting of outcomes (e.g. only positivity, or also bacterial counts).”

3.Conclusion states “reinfusion… is not consistently associated with adverse outcomes”, but many studies lacked power to detect infection difference.

We agree that many studies lacked power to detect difference. Therefore several lines were added to highlight this:

“On the other hand, it has to be mentioned that the studies included in this review lacked the statistical power to properly detect an increased risk of infection.”

“and generally insufficient sample sized to detect rare adverse events” was added in the sentence “However, substantial… definitive conclusions.” (Line 580)

“with sufficient sample size to detect adverse events.”

Reviewer #2:

Cell Salvage in Bacterially Contaminated Surgical Fields – a Scoping Review

I believe the summary could provide more definitive information on the subject, even though it remains controversial.

The introduction is lengthy, although very well-written, but could be adjusted to engage the reader.

The methods are very clear, with precise information. Additionally, the details are adequate, as are the inclusion and exclusion criteria.

The results are adequate, and the tables are very informative. The division of strategies by type of surgery seems quite appropriate.

The discussion and conclusion are well-founded and seem very appropriate to me.

I missed references and approaches to other surgeries, such as cesarean sections.

Cell Salvage in Bacterially Contaminated Surgical Fields – a Scoping Review

I believe the summary could provide more definitive information on the subject, even though it remains controversial.

We would like to thank the reviewer for the critical appraisal.

Considering the summary, we want to provide an overview of clinical evidence with our scoping review. We therefore hesitate to draw to firm a conclusion. But we did modify the summary slightly to draw a more firm conclusion than we did previously.

The introduction is lengthy, although very well

written, but could be adjusted to engage the reader.

We have carefully read and re-read the introduction section of the manuscript. We have carefully restructured it to make it more engaging.

I missed references and approaches to other surgeries, such as cesarean sections.

https://pubmed.ncbi.nlm.nih.gov/?term=+Intra-operative+cell+salvage

Thank you for pointing out the caesarean sections topic. We were also surprised that our search did not result in a higher yield regarding this type of surgery. However, this can be explained due to the in- and exclusion criteria of the study. A study was included whenever:

1) Studies had to examine or describe at least one of the following: measurement of bacterial contamination in salvaged blood, confirmed by blood cultures

2) bacterial reduction strategies implemented during the cell salvage process, such as leukocyte depletion filtration or antibiotic treatment of salvaged blood

3) or evaluation of clinical outcomes following reinfusion of bacterially contaminated salvaged blood.

In the appended search we have restricted it to caesarean sections:

https://pubmed.ncbi.nlm.nih.gov/?term=%28Intra-operative+cell+salvage%29+AND+%28%28sect%2A%29+OR+%28caes%2A%29%29&ac=no&sort=relevance

Which yielded n=15 articles. We have investigated the abstracts of the n=15 articles. Only one article was deemed appropriate for our scoping review, which is the article of Taere et al. Which is already present in our review.

Reviewer #3:

This is a well-conducted scoping review on intraoperative cell salvage (ICS) in bacterially contaminated surgical fields. The topic is highly relevant to clinical practice, and the authors followed established methodological frameworks (JBI, PRISMA-ScR). The manuscript is comprehensive and clearly structured, but some revisions would further strengthen its clarity, methodological rigor, and clinical impact.

While this is a scoping review rather than a systematic review, a brief assessment of study quality or risk of bias would help contextualize the findings. For example, highlighting differences between RCTs, prospective observational studies, and ex vivo studies could provide readers with a clearer sense of evidence strength.

In the Discussion section (around lines 343–369), where the authors discuss that “bacterial contamination does not necessarily lead to infection, possibly because low-level contamination can be cleared by the immune system,” they add the following:“Similar methodological challenges are seen in other clinical fields, where advanced statistical models have been applied to physiological signals to identify subtle but clinically relevant features (cite: Applied statistical methods for identifying features of heart rate that are associated with nicotine vaping).”

Please ensure consistent use of terms: “salvaged red blood cells (sRBCs)” vs. “ICS blood.”

Standardize abbreviations (e.g., ICS, LDF, CFU/mL) across text, tables, and figures.

Parts of the discussion (e.g., lines 343–369) contain repeated sentence structures such as “was found to” and “was reported to.” Rephrasing for variety would improve readability.

The conclusion could be more concise, emphasizing key clinical take-home messages for practitioners.

While this is a scoping review rather than a systematic review, a brief assessment of study quality or risk of bias would help contextualize the findings. For example, highlighting differences between RCTs, prospective observational studies, and ex vivo studies could provide readers with a clearer sense of evidence strength

We would like to thank the reviewer for the critical appraisal. Methodological quality assessment is not usual in scoping reviewers. However, as two of three reviewers mentioned assessing methodological quality we have performed a quality assessment using two tools. The Risk of Bias 2 tool for the Randomized controlled trials and the Newcastle-Ottawa Scale assessment for cohort studies included in this review.

“Methodological quality

The quality assessment of articles included was carried out by JS and PH independently. Whenever applicable a third reviewer (AK) arbitrated. The revised Cochrane risk of bias 2 tool for randomised controlled trials (RoB 2) was used for the included randomised controlled trials [22]. This tool uses stratification into five domains to detect potential bias. For included articles other than randomised controlled trials the Newcastle-Ottawa Score was used [23]. This tool uses stratification into three domains to score quality. Ex-vivo studies were not assessed for methodological quality.” was added in the methodology section of the paper.

“Table 4 and Figure 2 summarize the quality assessments of RCTs (n=4) and cohort studies (n=31), except for the ex-vivo studies. The RCTs were assessed as some concerns (n=4). The cohort studies were assessed as poor (n=28) or good (n=3).” was added in the results section.

A table 4 figure 2 were added in the main paper below table 3.

In the Discussion section (around lines 343–369), where the authors discuss that “bacterial contamination does not necessarily lead to infection, possibly because low

level contamination can be cleared by the immune system,” they add the following:“Similar methodological challenges are seen in other clinical fields, where advanced statistical models have been applied to physiological signals to identify subtle but clinically relevant features (cite: Applied statistical methods for identifying features of heart rate that are associated with nicotine vaping).”

Thank you for the suggestion. We have added your suggestion to our discussion section following the sentence which you mentioned. We also added the reference.

Please ensure consistent use of terms: “salvaged red blood cells (sRBCs)” vs. “ICS blood.”

Standardize abbreviations (e.g., ICS, LDF, CFU/mL) across text, tables, and figures.

Thank you for highlighting the inconsistency between sRBCs and ICS blood. We have changed this uniformly to “ICS blood” throughout the document, inconsistencies appeared predominantly in table 3 and the results section.

Thank you we have checked our tables on the used abbreviations and standardized the abbreviations.

Parts of the discussion (e.g., lines 343–369) contain repeated sentence structures such as “was found to” and “was reported to.” Rephrasing for variety would improve readability.

Yes indeed. Thank you for noticing, we used the term “found” in approximately six sentences. In three sentences, where appropriate, we altered “found” for “reported” to improve variety and readability throughout the document.

The conclusion could be more concise, emphasizing key clinical take

home messages for practitioners.

Considering your feedback on the conclusion to be more concise. That is difficult to establish. Reviewer #1 mentioned that the missing ‘causal relationship’ and missing ‘safe contamination threshold’ limits strength in the conclusion. Reviewer #2 found the discussion well-founded and appropriate. As we are not able to describe causal relationship or define contamination thresholds we had to contextualize the conclusion, resulting in extra information.

---

## [Decision Letter · Decision Letter 1]

9 Dec 2025

Cell Salvage in Bacterially Contaminated Surgical Fields – a Scoping Review

PONE-D-25-38734R1

Dear Dr. Koopman,

We’re pleased to inform you that your manuscript has been judged scientifically suitable for publication and will be formally accepted for publication once it meets all outstanding technical requirements.

Kind regards,

Miquel Vall-llosera Camps

Senior Staff Editor

PLOS One

Reviewers' comments:

Reviewer's Responses to Questions

**Comments to the Author**

Reviewer #2: (No Response)

Reviewer #3: All comments have been addressed

2. Is the manuscript technically sound, and do the data support the conclusions?

Reviewer #2: Yes

Reviewer #3: Yes

3. Has the statistical analysis been performed appropriately and rigorously?

Reviewer #2: Yes

Reviewer #3: Yes

4. Have the authors made all data underlying the findings in their manuscript fully available?

Reviewer #2: Yes

Reviewer #3: Yes

5. Is the manuscript presented in an intelligible fashion and written in standard English?

Reviewer #2: Yes

Reviewer #3: Yes

Reviewer #2: Dear authors and editors,

All requests have been fulfilled and I believe the article is now suitable for publication in Plos One.

Reviewer #3: The authors have adequately addressed all concerns raised in the previous round of review. The revised manuscript is clearly structured, methodologically appropriate, and written in fluent, standard English. The data extraction methods and inclusion criteria are transparent, and the conclusions are well supported by the presented evidence. I have no further comments or suggestions. I recommend the manuscript for acceptance.

**Do you want your identity to be public for this peer review?** For information about this choice, including consent withdrawal, please see our Privacy Policy

Reviewer #2: **Yes: ** Vicente Sperb Antonello

Reviewer #3: No

---

## [Editor Report · Acceptance letter]

PONE-D-25-38734R1

PLOS One

Dear Dr. Koopman,

I'm pleased to inform you that your manuscript has been deemed suitable for publication in PLOS One. Congratulations! Your manuscript is now being handed over to our production team.

Kind regards,

on behalf of

Dr. Miquel Vall-llosera Camps

Staff Editor

PLOS One